# Simulating Chern insulators on a superconducting quantum processor

Zhong-Cheng Xiang[1,10], Kaixuan Huang [1,2,3,10], Yu-Ran Zhang [4,5,6,10], Tao Liu[4], Yun-Hao Shi [1], Cheng-Lin Deng[1], Tong Liu [1], Hao Li [1], Gui-Han Liang[1], Zheng-Yang Mei[1], Haifeng Yu [2], Guangming Xue[2], Ye Tian[1], Xiaohui Song[1], Zhi-Bo Liu[3], Kai Xu [1,2,7,8] ✉, Dongning Zheng [1,7,8], Franco Nori [5,6,9] ✉ & Heng Fan [1,2,7,8] ✉

The quantum Hall effect, fundamental in modern condensed matter physics, continuously inspires new theories and predicts emergent phases of matter. Here we experimentally demonstrate three types of Chern insulators with synthetic dimensions on a programable 30-qubit-ladder superconducting processor. We directly measure the band structures of the 2D Chern insulator along synthetic dimensions with various configurations of Aubry-André-Harper chains and observe dynamical localisation of edge excitations. With these two signatures of topology, our experiments implement the bulk-edge correspondence in the synthetic 2D Chern insulator. Moreover, we simulate two different bilayer Chern insulators on the ladder-type superconducting processor. With the same and opposite periodically modulated on-site potentials for two coupled chains, we simulate topologically nontrivial edge states with zero Hall conductivity and a Chern insulator with higher Chern numbers, respectively. Our work shows the potential of using superconducting qubits for investigating different intriguing topological phases of quantum matter.

Topological phases of matter[1,2], classified beyond Landau's symmetry-breaking theory, have been attracting growing interest in recent decades. It started with the discovery of the two-dimensional (2D) integer quantum Hall effect (QHE)[3], which arises from the topological nature of Bloch bands characterised by the Chern number[4]. Topological band theory provides a direct link between theory and experiments, which are successful for identifying salient characteristics of topological states and predicting new classes of topological phases[5]. The existence of robust edge states is deeply related to the topology of gapped bulk band structures, which is the ubiquitous bulk-edge correspondence in topological systems.

Since exploring higher-dimensional physics requires an exponentially growing number of qubits, there is growing interest in creating synthetic dimensions to construct a higher-dimensional lattice in a lower-dimensional system with its internal degrees of freedom[6–9]. Moreover, a lower-dimensional topological charge pump shares the same topological origin as higher-dimensional topological physics, e.g., a one-dimensional (1D) Thouless pump for the 2D integer QHE[10] and a 2D topological pump for a four-dimensional quantum Hall system[11,12].

In addition to the 2D electron gas, the integer QHE and Chern insulators (i.e., a lattice version of the QHE) have also been observed

[1]Institute of Physics, Chinese Academy of Sciences, Beijing 100190, China. [2]Beijing Academy of Quantum Information Sciences, Beijing 100193, China. [3]Key Laboratory of Weak Light Nonlinear Photonics, Ministry of Education, Teda Applied Physics Institute and School of Physics, Nankai University, Tianjin 300457, China. [4]School of Physics and Optoelectronics, South China University of Technology, Guangzhou 510640, China. [5]Theoretical Quantum Physics Laboratory, Cluster for Pioneering Research, RIKEN, Wako-shi, Saitama 351-0198, Japan. [6]Center for Quantum Computing, RIKEN, Wako-shi, Saitama 351-0198, Japan. [7]CAS Centre for Excellence in Topological Quantum Computation, UCAS, Beijing 100190, China. [8]Songshan Lake Materials Laboratory, Dongguan 523808, China. [9]Physics Department, University of Michigan, Ann Arbor, MI 48109-1040, USA. [10]These authors contributed equally: Zhong-Cheng Xiang, Kaixuan Huang, Yu-Ran Zhang. ✉e-mail: kaixu@iphy.ac.cn; fnori@riken.jp; hfan@iphy.ac.cn

in other physical platforms, including ultra-cold atoms in optical lattices[13–15], photonic systems[16–18], etc.[19,20]. However, it still remains very challenging to synthesise topological quantum phases and to further demonstrate the bulk-edge correspondence in quantum simulation platforms[21–23].

Here we observe several topological signatures of 2D and bilayer Chern insulators with synthetic dimensions on a programmable 30-qubit-ladder superconducting processor. We experimentally measure the band structure of the 2D Chern insulator along a synthetic dimension by analysing the temporal frequency of the system's response to local perturbations. By monitoring quantum walks (QWs) of a single excitation initially prepared at an edge qubit, we demonstrate dynamical localisation of the topologically protected edge states. The measured band structure and the dynamical signatures of topological edge states together demonstrate the bulk-edge correspondence. Furthermore, we synthesise two different bilayer Chern insulators on the 30-qubit-ladder processor. Given the same periodically modulated on-site potentials of two coupled chains, we probe the nontrivial topological edge states with zero Hall conductivity. When on-site potentials of two coupled chains have opposite signs, a Chern insulator with higher Chern numbers is probed. Our results show that superconducting simulation platforms are capable for studying different intriguing topological phases of quantum matter.

## Results

### Experimental setup and model Hamiltonians

Our experiments are performed on a superconducting circuit[24] consisting of 30 transmon qubits ($Q_{j,s}$, with $j$ varied from 1 to 15 and pseudo-spin $s \in \{\uparrow, \downarrow\}$)[25,26], which constitute a two-legged qubit ladder[27], see Fig. 1a. By setting $\hbar = 1$, the system's Hamiltonian can be written as[28–30]

$$H = J_\parallel \sum_{j,s}(\hat{c}_{j,s}^\dagger \hat{c}_{j+1,s} + \text{H.c.}) + J_\perp \sum_j(\hat{c}_{j,\uparrow}^\dagger \hat{c}_{j,\downarrow} + \text{H.c.}) \\ + \sum_{j,s} V_{j,s}\hat{c}_{j,s}^\dagger \hat{c}_{j,s}, \quad (1)$$

where $\hat{c}^\dagger$ ($\hat{c}$) is the hard-core bosonic creation (annihilation) operator with $(\hat{c}^\dagger)^2 = \hat{c}^2 = 0$, and $[\hat{c}_{i,s}^\dagger, \hat{c}_{j,r}] = \delta_{ij}\delta_{sr}$. Here, $J_\parallel/2\pi \simeq 8$ MHz and

$J_\perp/2\pi \simeq 7$ MHz denote the nearest-neighbour (NN) hopping between nearby qubits on the same leg and on the same rung, respectively, and $V_{j,s}$ is the tunable on-site potential. Experimental details of our system are described in the Supplementary Note 1 and Supplementary Note 2.

With a dimensional reduction procedure[31], a 2D integer quantum Hall system can be mapped to a 1D model with a periodic parameter as a synthetic dimension. In this context, we experimentally simulate a 2D integer QHE associated with Chern insulator using 15 qubits on one leg of the ladder (i.e., qubits labelled by $Q_{1,\uparrow}$ to $Q_{15,\uparrow}$), where the on-site potential $V_{j,\uparrow}$ of each qubit is periodically modulated. This system can be described by the 1D Aubry-André-Harper (AAH) model[32,33] with a tight-binding Hamiltonian

$$H_{AAH} = J_\parallel \sum_{j=1}^{14}(\hat{c}_j^\dagger \hat{c}_{j+1} + \text{H.c.}) + \Delta \sum_{j=1}^{15}\cos(2\pi bj + \phi)\hat{c}_j^\dagger \hat{c}_j, \quad (2)$$

where the second index $\uparrow$ is omitted. Here, $b$ determines the modulation periodicity, and the modulation phase $\phi$ corresponds to the momentum in a synthetic dimension[31], see Fig. 1b. Note that there exists hopping between next-nearest neighbour (NNN) qubits $H' = J'_\parallel \sum_{j=1}^{13}(\hat{c}_j^\dagger \hat{c}_{j+2} + \text{H.c.})$ on the same leg with a strength $J'_\parallel \simeq 0.1 J_\parallel$, see the Supplementary Note 1. This model is topologically equivalent to the lattice model of the 2D integer QHE, proposed by Hofstadter[34], where electrons hop within the 2D lattice, subjected to a perpendicular magnetic field with $b$ being the magnetic flux (normalised to the magnetic flux quantum) threading each unit cell. In our experiments, we fix $b = \frac{1}{3}$, and vary $\phi$ from 0 to $2\pi$ to obtain various instances of AAH models with potentials by tuning the qubits' frequencies as $\omega_j = \omega_0 + \Delta \sum_{j=1}^{15}\cos(2\pi bj + \phi)$, with a reference frequency $\omega_0/2\pi = 4.7$ GHz (Fig. 1c).

### Band structure spectroscopy

Band structures play an essential role in characterising topological phases of matter and discovering novel classes of intriguing topological materials[5]. Here, we directly measure the topological band structure of the integer quantum Hall system along the synthetic dimension using a dynamic spectroscopic technique applied in refs. 28,35,36. This method detects quantised eigenenergies of

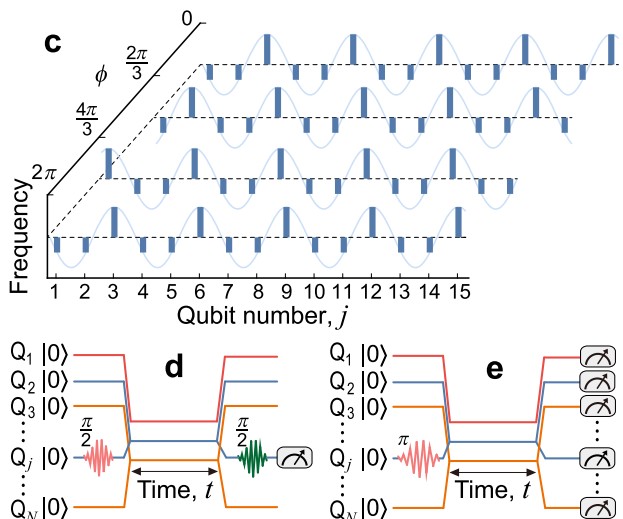

**Fig. 1 | Quantum simulation of Chern insulators on a 30-qubit-ladder superconducting processor. a** Schematic of the superconducting quantum processor, where 30 superconducting qubits constitute a ladder. Each qubit, coupled to an independent readout resonator (R), has an independent microwave line for XY and Z controls. **b** Mapping the 2D Hofstadter model to various configurations of Aubry-

André-Harper (AAH) chains with a Fourier transformation (FT) along the $y$-axis with $M$ sites. **c** Qubits' frequencies for the synthesis of a series of AAH chains with different values of $\phi$ and $b = \frac{1}{3}$. **d, e** Experimental waveform sequences for the dynamic band structure spectroscopy (**d**) and the single-particle quantum walks (QWs) (**e**).

the quantum system from the Fourier transformation (FT) of the subsequent response of the system given local perturbations. The experimental sequence of pulses of the band structure spectroscopy are shown in Fig. 1d. With 15 qubits initialised at their idle points, we place one target qubit $Q_j$ in the superposed state $|+\rangle_j = (|0_j\rangle + |1_j\rangle)/\sqrt{2}$, using a $Y_{\frac{\pi}{2}}$ pulse. Then, all qubits are detuned to their corresponding frequencies for the quench dynamics with a time $t$ before the readout of the $Q_j$ at its idle point in the $\hat{\sigma}^x$ and $\hat{\sigma}^y$ bases. For each $\phi$, time evolutions of $\langle\hat{\sigma}_j^x\rangle$ and $\langle\hat{\sigma}_j^x\rangle$ are recorded when choosing a target qubit, e.g., $Q_8$ in Fig. 2a. Figure 2b shows the squared FT magnitude $|A_j|^2$ of the response function $\chi_j(t) \equiv \langle\hat{\sigma}_j^x(t)\rangle + i\langle\hat{\sigma}_j^y(t)\rangle$ for each qubit with $\phi = \frac{2\pi}{3}$ and $\Delta/2\pi = 12$ MHz. With the summation of the squared FT magnitudes of all selected qubits $I_\phi \equiv \sum_j |A_j|^2$, the positions of its peaks clearly indicate the eigenenergies $E/2\pi$ of the system for each $\phi$ (Fig. 2c).

### Simulating 2D Chern insulators with a synthetic dimension

The topological nature of the 1D AAH model and the 2D integer QHE associated with Chern insulator can be identified from its band structure. When setting $\Delta/2\pi = 12$ MHz, we plot in Fig. 2d the band structure of the 1D AAH model along $\phi$ with $N = 15$ and open boundary conditions along $x$-direction. As $\phi$ evolves, the gapless edge states (red curves) appear in two gaps between three "bulk band" regimes (dense blue curves). We experimentally map out this band structure by measuring $I_\phi$ for $\phi \in [0, 2\pi]$ (Fig. 2e), which also agrees well with the numerical result by simulating the system dynamics (Fig. 2f). Two gapless Dirac bands within two band gaps are clearly observed in Fig. 2e, and each gap is related to a quantised Hall conductance $\sigma = (e^2/h)\mathcal{C}$, with the integer $\mathcal{C}$ being determined by the Chern number[4].

Moreover, the topological edge states predicted in the band structure can be verified in real space by observing localisation of an edge excitation during its QWs on the 15-qubit chain[37]. After the system initialisation, we excite one qubit with a $X_\pi$ pulse, tune all qubits to their corresponding frequencies and measure them at a time $t$ after their free evolutions (Fig. 1e). For the topologically trivial case with $\Delta/2\pi = 0$ MHz, the measured density distributions $P_j(t)$ of single-excitation QWs initialised at any position of the qubit chain show a light-cone-like propagation of information with boundary reflections[30].

When we set $b = \frac{1}{3}$ and $\Delta/2\pi = 12$ MHz, the $P_j(t)$ for the excitation initialised at a boundary qubit ($Q_1$ or $Q_{15}$) exhibits localisation for $\phi = \frac{2\pi}{3}$, where the edge states appear in the middle of a band gap (Fig. 3a1, a3). This localised dynamical behaviour is due to the fact that the edge-excitation modes have a main overlap with the in-gap edge states which are topologically protected by the band gaps. This is also verified by the squared FT magnitudes $|A_1|^2$ and $|A_{15}|^2$ for edge qubits (Fig. 3b1, b3), which mainly contain information of the in-gap edge states. The discontinuity of the FT signals for edge qubits results from the second and first band gaps, respectively, because the edge states localised at two boundaries have opposite propagation directions, like in the standard QHE. When a qubit away from either end, e.g. $Q_8$, is excited, we observe the propagation of the excitation in Fig. 3a2 due to the absence of topological protection, and its FT signal $|A_8|^2$ merely shows partial information of the bulk band (Fig. 3b2). Thus, by directly measuring the band structure and observing dynamical localisation of edge excitations, our experiments demonstrate the bulk-edge correspondence in the synthetic 2D Chern insulator.

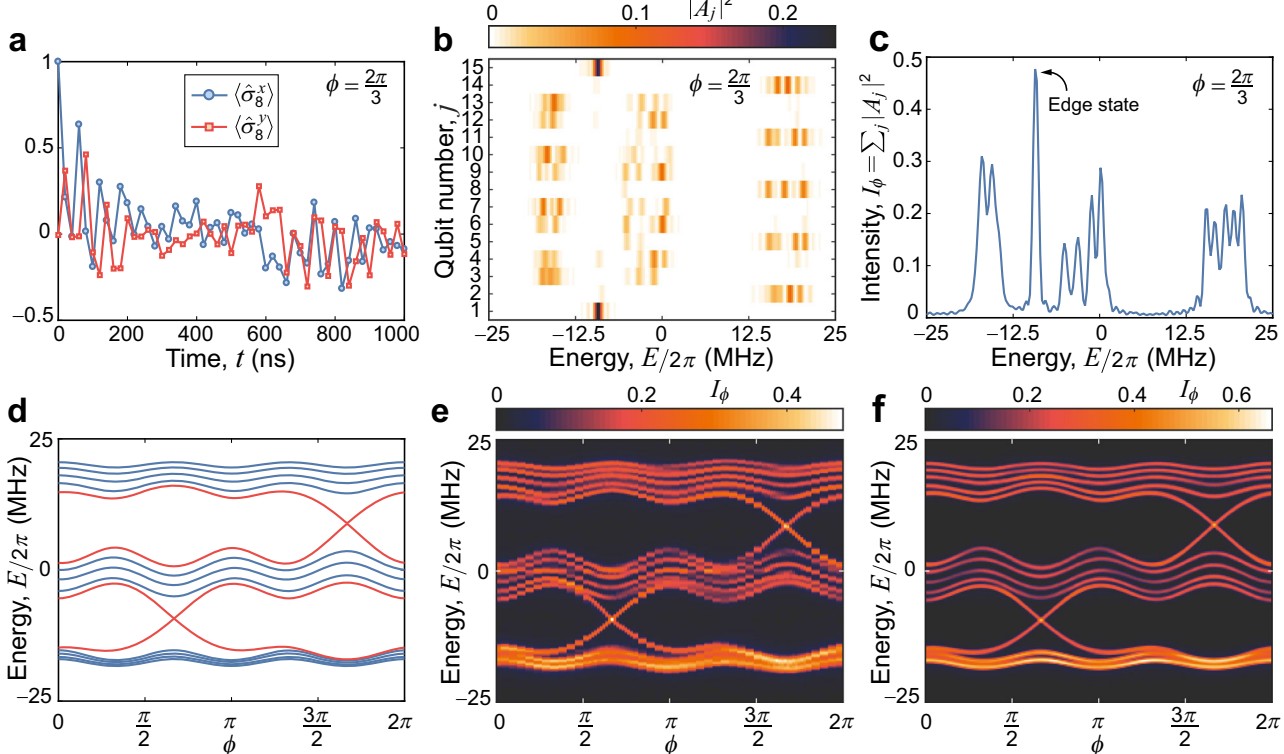

**Fig. 2 | Band structure spectroscopy of the 2D Chern insulator with a synthetic dimension. a** Typical data of $\langle\hat{\sigma}^x\rangle$ and $\langle\hat{\sigma}^y\rangle$ versus time $t$ when choosing $Q_8$ as the target qubit for $b = \frac{1}{3}$, $\Delta/2\pi = 12$ MHz and $\phi = \frac{2\pi}{3}$. **b** Squared FT magnitudes $|A_j|^2$ of the response functions $\chi_j(t) \equiv \langle\hat{\sigma}_j^x(t)\rangle + i\langle\hat{\sigma}_j^y(t)\rangle$ for all fifteen qubits. Note that the edge states at the edge qubits: $Q_1$ and $Q_{15}$. **c** Summation of the squared FT magnitudes $I_\phi \equiv \sum_j |A_j|^2$. **d** Band structure of the 2D Chern insulator for $b = \frac{1}{3}$ and $\Delta = 12$ MHz with

15 sites along the $x$-direction and periodic along the $y$ direction. Red curves show the edge states. **e, f** Experimentally measured data for $I_\phi$ (**e**) for different values of $\phi$ varied from 0 to $2\pi$ are compared with numerically calculated data of $I_\phi$ (**f**) obtained by numerically simulating the dynamics of the 15-qubit system without decoherence.

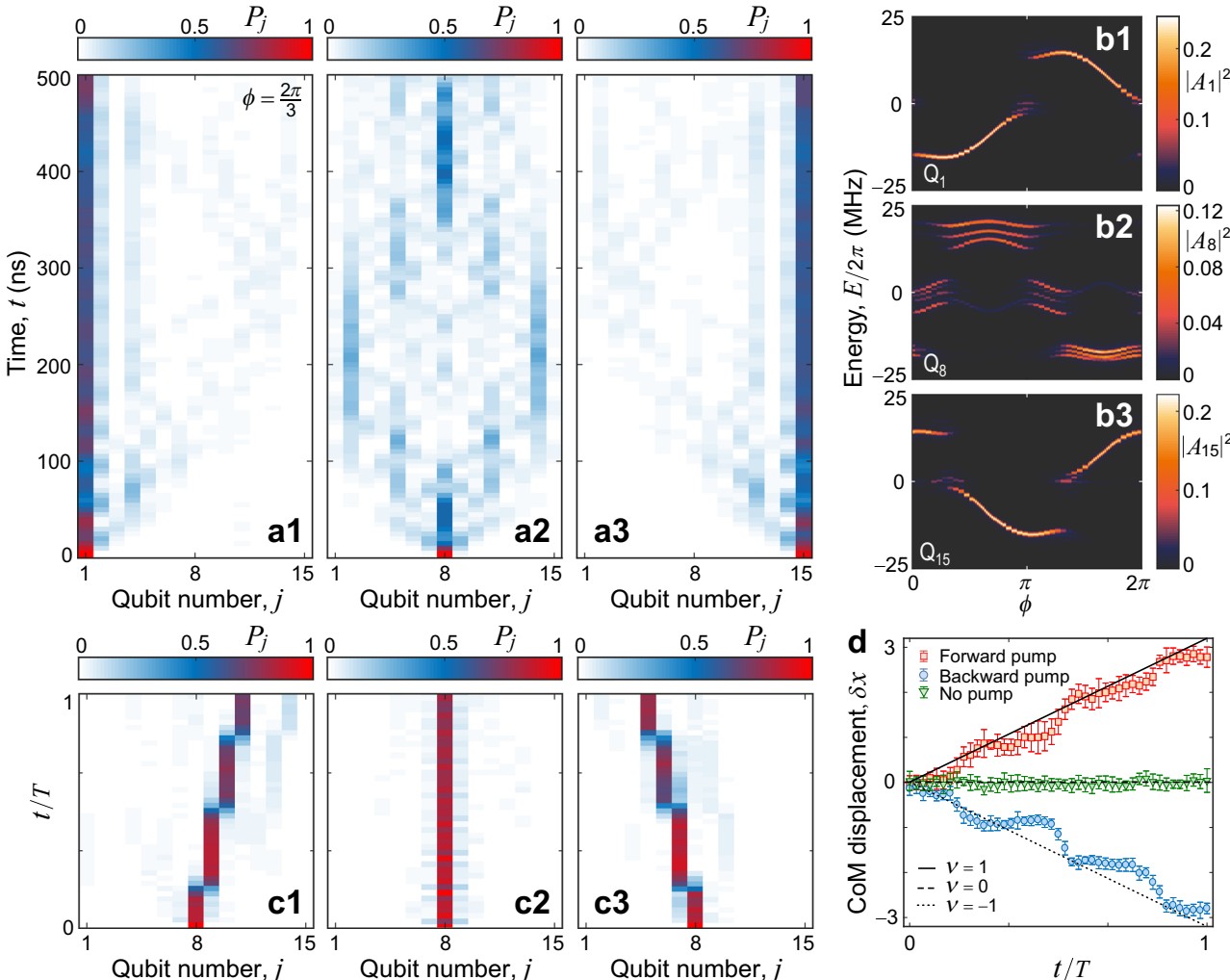

**Fig. 3 | Dynamical signatures of topological edge states and the topological charge pump. a1–a3** Time evolution of the excitation probability $P_j$ with $b = \frac{1}{3}$, $\Delta/2\pi = 12$ MHz and $\phi = 2\pi/3$ after initially exciting the leftmost qubit $Q_1$ (**a1**), the central qubit $Q_8$ (**a2**), and the rightmost qubit $Q_{15}$ (**a3**). **b1–b3,** Experimental data of the squared FT magnitudes $|A_1|^2$, $|A_8|^2$, and $|A_{15}|^2$, when choosing $Q_1$ (**b1**), $Q_8$ (**b2**), and $Q_{15}$ (**b3**) as the target qubits. **c1–c3,** Time evolution of an excitation initially prepared at the central qubit $Q_8$ when it is forward pumped (**c1**), not pumped (**c2**) and backward pumped (**c3**), respectively, with $\Delta/2\pi = 36$ MHz for an initial $\phi_0 = 5\pi/3$. **d** Displacement of the centre of mass (CoM) $\delta x$ versus time $t$ in one pumping cycle with period $T$ for the cases in (**c1–c3**). The error bars are 1 SD, calculated from all 10 groups of experimental results.

In addition, a topological charge pump, entailing the charge transport in a 1D time-varying potential driven in adiabatic cycles[10,38], provides an alternative way to explore the 2D integer QHE. The charge transported in a pumping cycle is determined by the Chern number[39], which is defined over a 2D Brillouin zone with one spatial and one temporal dimensions. We experimentally simulate the charge pump by adiabatically varying $\phi$ in a $2\pi$ period starting from $\phi_0 = \frac{5\pi}{3}$ with $\Delta/2\pi = 36$ MHz. Figure 3c1, c3 plot the evolutions of distributions $P_j(t)$ of an excitation initialised at the central qubit $Q_8$ for the forward and backward pumping schemes, respectively. In one pumping cycle, the excitation propagates through an integer number of unit cells, in this case three qubits, which is determined by the Chern number.

Following Laughlin's argument[40,41], the role of the threading magnetic flux is played by the adiabatic variation of $\phi$, leading to the excitation's transport. The displacements of the centre of mass (CoM) $\delta x$ are shown in Fig. 3d, of which the slight deviations from the Chern numbers $\pm 1$ may result from the boundary reflection on the finite-size 1D qubit chain and the small fraction of excitations for the higher bands. A brief discussion of fast pumping[42] is given in the Supplementary Note 3. In comparison,

there exists no excitation transport when $\phi$ is not pumped (Fig. 3c2, d).

## Simulating bilayer Chern insulators

Next, using all thirty qubits, we simulate two different bilayer Chern insulators. The Hamiltonian of the 30-qubit-ladder quantum processor is given in equation (1), where the on-site potentials on the qubits in two legs ($s \in \{\uparrow, \downarrow\}$) are modulated as

$$V_{j,s} = \Delta_s \cos(2\pi b j + \phi). \tag{3}$$

with $b = \frac{1}{3}$. For instances of two coupled AAH chains by varying $\phi$ with the same $\Delta_{\uparrow(\downarrow)}/2\pi = 12$ MHz and opposite $\Delta_\uparrow/2\pi = -\Delta_\downarrow/2\pi = 12$ MHz, we effectively simulate two different bilayer quantum systems[43], where the magnetic fluxes threading two layers are the same and have a difference of $\pi$, respectively. Using the above spectroscopic technique, we measure the band structures for these two cases (Fig. 4a, b), which agree well with the theoretical predictions (dashed curves) using the experimental parameters in the Supplementary Note 1.

For $\Delta_{\uparrow(\downarrow)}/2\pi = 12$ MHz with $J_\perp$ being comparable to $J_\parallel$, gapless edge states are experimentally observed (Fig. 4a), even when the

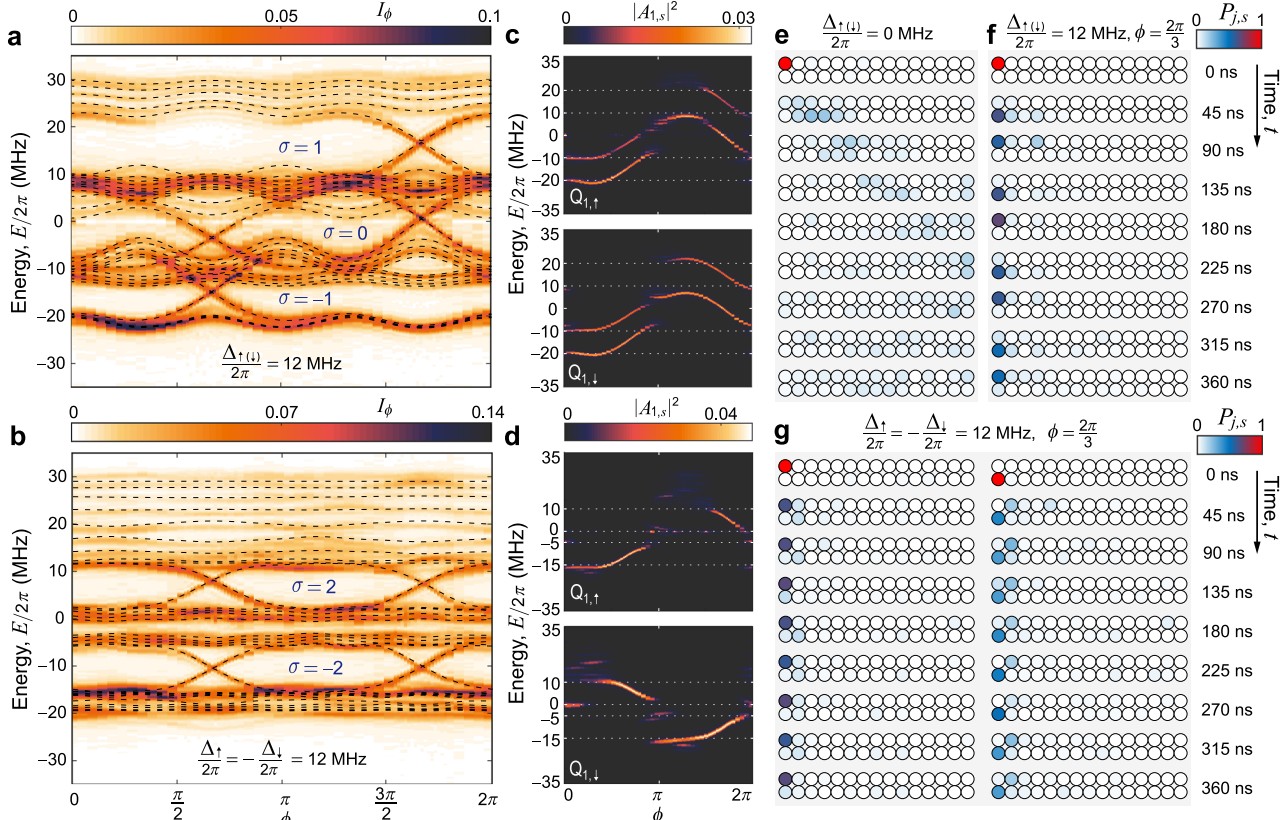

**Fig. 4 | Quantum simulation of bilayer Chern insulators using all thirty qubits on a ladder-type quantum processor. a**, **b** Experimentally measured $I_\phi$ with $b = \frac{1}{3}$ versus $\phi$ varied from 0 to $2\pi$ for two AAH chains with the same on-site potentials $\Delta_{\uparrow(\downarrow)}/2\pi = 12$ MHz (**a**) and opposite on-site potentials $\Delta_\uparrow/2\pi = -\Delta_\downarrow/2\pi = 12$ MHz (**b**), respectively, which are compared with the theoretical projected band structures (dashed curves). The Hall conductivity is defined as the summation of the Chern number $\mathcal{C}_n$ over the occupied bands: $\sigma = \sum_n \mathcal{C}_n$ by letting $e^2/h = 1$. **c**, **d** Experimental data of the squared FT magnitudes $|A_{1,\uparrow}|^2$ and $|A_{1,\downarrow}|^2$ when choosing $Q_{1,\uparrow}$ and $Q_{1,\downarrow}$ as the target qubits for $\Delta_{\uparrow(\downarrow)}/2\pi = 12$ MHz (**c**) and $\Delta_\uparrow/2\pi = -\Delta_\downarrow/2\pi = 12$ MHz (**d**), respectively. **e**, **g** Time evolutions of the excitation probability $P_{j,s}$, with $\phi = \frac{2\pi}{3}$, after initially exciting a corner qubit ($Q_{1,\uparrow}$ or $Q_{1,\downarrow}$) for $\Delta_{\uparrow(\downarrow)}/2\pi = 0$ MHz (**e**), $\Delta_{\uparrow(\downarrow)}/2\pi = 12$ MHz (**f**), and $\Delta_\uparrow/2\pi = -\Delta_\downarrow/2\pi = 12$ MHz (**g**), respectively. Animations of the time evolutions are available in Supplementary Movie 1.

Chern number for half filling is predicted to be zero (see Methods). This novel topological phase results from the existence of a pair of counter-propagating chiral edge states in the second gap (Fig. 4c), which are protected by the inversion symmetry and lead to a zero Hall conductivity $\sigma = 0$[44]. When we excite a corner qubit (e.g., $Q_{1,\uparrow}$) and monitor the excitation's QWs for $\phi = \frac{2\pi}{3}$, the time-evolved distribution $P_{j,s}$ shows oscillations between two corner states localised at the same boundary rung of the qubit ladder (Fig. 4f). This dynamical behaviour can be further understood from the measured squared FT magnitudes $|A_{1,\uparrow}|^2$ and $|A_{1,\downarrow}|^2$ with respect to the corner qubits $Q_{1,\uparrow}$ and $Q_{1,\downarrow}$, respectively, which share information of the same edge states (Fig. 4c).

By modulating opposite on-site potential fields with $\Delta_\uparrow/2\pi = -\Delta_\downarrow/2\pi = 12$ MHz, we obtain a different band structure (Fig. 4b) from the one with two layers having the same magnetic flux. In this case, the magnetic fluxes threading two layers have a difference of $\pi$, and the gapless edge states are characterised by higher Chern numbers (see Methods). The QWs of the excitation initialised at a corner qubit (e.g., $Q_{1,\uparrow}$ or $Q_{1,\downarrow}$) for $\phi = \frac{2\pi}{3}$ show dynamical localisation at the initial qubit (Fig. 4g), and the measured squared FT magnitudes of two corner qubits show information of different edge states in different gaps (Fig. 4d). Thus, this synthetic bilayer quantum system with two layers having a $\pi$-flux difference presents a Chern insulator with chiral edge states, identified by higher Chern numbers. Since high-Chern-number insulators without the formation of Landau levels have attracted increasing attentions[45–47], our work provides a new perspective for exploring these emergent topological phases. In comparison, for $\Delta_{\uparrow(\downarrow)}/2\pi = 0$, the QWs of the corner excitation first show linear propagation and then indicate thermalisation on the qubit ladder as a non-integrable system[29,48].

## Discussion

In summary, we simulate 2D and bilayer Chern insulators with synthetic dimensions on a programmable 30-qubit-ladder superconducting processor. By measuring the band structures and monitoring dynamical localisation of edge excitations, we implement the bulk-edge correspondence in the 2D Chern insulator. In addition, we synthesise two different bilayer Chern insulators with two layers having the same magnetic flux and a $\pi$-flux difference, respectively, and simulate distinct and novel topological phases. Our experiments, using a relatively large number of superconducting qubits with long coherence times and accurate readouts, show the future potential of using superconducting simulating platforms for investigating intriguing topological quantum phases and quantum many-body physics. For instance, by upgrading the ladder-type superconducting processor with tunable interactions between two qubit chains, we could observe the quantum phase transitions between topological phases with zero Chern number (Fig. 5b) and with higher Chern numbers (Fig. 5c). In addition, when tuning weak couplings between two chains and modulating $b$ either positive or negative for two chains, respectively, the quantum spin Hall effect could be

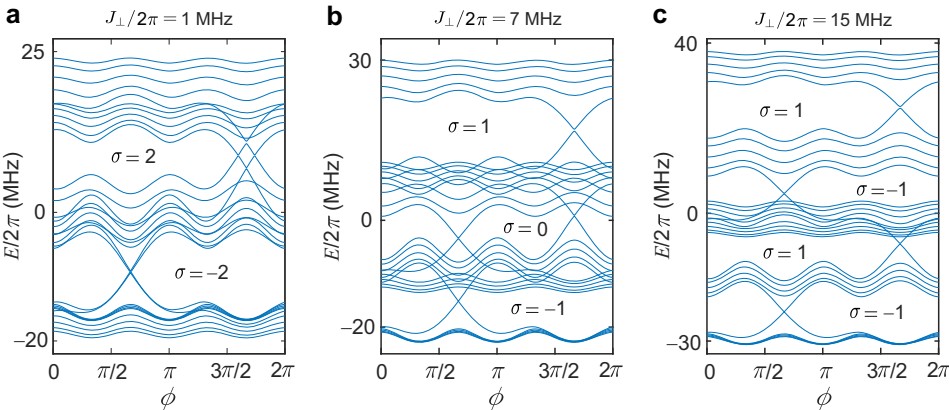

**Fig. 5 | Energy spectra for the bilayer topological system with the same periodically modulated on-site potentials. a–c** Energy spectra versus $\phi$ of a finite ladder with 30 sites for the same modulated amplitudes $\Delta_\uparrow/2\pi = \Delta_\downarrow/2\pi = 12$ MHz with $J_\perp/2\pi = 1$ MHz (**a**), $J_\perp/2\pi = 7$ MHz (**b**), and $J_\perp/2\pi = 15$ MHz (**c**). Other parameters used are $J_\parallel/2\pi = 8$ MHz, $J'_\parallel = 0.1J_\parallel$, and $J_\times = 0.2J_\parallel$.

demonstrated by simulating bilayer quantum Hall systems with opposite Chern numbers[49].

## Methods
### System Hamiltonian
Our superconducting quantum processor can be described as a Bose-Hubbard ladder with a Hamiltonian ($\hbar = 1$)[28,29]

$$H_{\mathrm{BH}} = J_\parallel \sum_{j,s} (\hat{a}^\dagger_{j,s}\hat{a}_{j+1,s} + \mathrm{H.c.}) + \sum_{j,s} \frac{\eta_{j,s}}{2}\hat{n}_{j,s}(\hat{n}_{j,s} - 1)$$
$$+ J_\perp \sum_j (\hat{a}^\dagger_{j,\uparrow}\hat{a}_{j,\downarrow} + \mathrm{H.c.}) + \sum_{j,s} V_{j,s}\hat{n}_{j,s}, \qquad (4)$$

where $\hat{a}^\dagger$ ($\hat{a}$) is the bosonic creation (annihilation) operator, and $\hat{n} \equiv \hat{a}^\dagger\hat{a}$ is the number operator. Here, $J_\parallel/2\pi \simeq 8$ MHz and $J_\perp/2\pi \simeq 7$ MHz denote the nearest-neighbour (NN) hopping between nearby qubits on the same leg and on the same rung, respectively. Also, $\eta$ is the on-site nonlinear interaction, and $V_{j,s}$ is the tunable on-site potential.

Our device is designed to fulfil the hard-core limit $|\eta/J| \gg 1$, and thus, the highly occupied states of transmon qubits are blockaded, which represents fermionisation of strongly interacting bosons[30]. The system Hamiltonian can then be simplified as

$$H = J_\parallel \sum_{j,s} (\hat{c}^\dagger_{j,s}\hat{c}_{j+1,s} + \mathrm{H.c.}) + \sum_{j,s} V_{j,s}\hat{c}^\dagger_{j,s}\hat{c}_{j,s}$$
$$+ J_\perp \sum_j (\hat{c}^\dagger_{j,\uparrow}\hat{c}_{j,\downarrow} + \mathrm{H.c.}), \qquad (5)$$

where $\hat{c}^\dagger$ ($\hat{c}$) is the hard-core bosonic creation (annihilation) operator with $(\hat{c}^\dagger)^2 = \hat{c}^2 = 0$, and $[\hat{c}_{j,s}, \hat{c}_{i,r}] = \delta_{ji}\delta_{sr}$.

Note that in addition to the hopping between nearest-neighbour (NN) qubits, there also exist the hopping between next-nearest neighbour (NNN) qubits on different legs:

$$J_\times \sum_j (\hat{c}^\dagger_{j,\uparrow}\hat{c}_{j+1,\downarrow} + \hat{c}^\dagger_{j,\downarrow}\hat{c}_{j+1,\uparrow} + \mathrm{H.c.}), \qquad (6)$$

and the hopping between third-nearest-neighbour (TNN) qubits on the same leg:

$$J'_\parallel \sum_{j,s} (\hat{c}^\dagger_{j,s}\hat{c}_{j+2,s} + \mathrm{H.c.}). \qquad (7)$$

## Mapping the 2D quantum Hall model to various instances of Aubry-André-Harper chains
An electron moving in a 2D lattice with a perpendicular magnetic field $b$ is described by the integer quantum Hall model,

$$H_{\mathrm{IQH}} = t_x \sum_{j=1}^{N-1}\sum_{m=1}^{M} (\hat{c}^\dagger_{j,m}\hat{c}_{j+1,m} + \mathrm{H.c.})$$
$$+ t_y \sum_{j=1}^{N}\sum_{m=1}^{M} (\hat{c}^\dagger_{j,m}\hat{c}_{j,m+1}e^{i2\pi bj} + \mathrm{H.c.}), \qquad (8)$$

where $t_x$ and $t_y$ are hopping strengths along the $x$ and $y$-axes, respectively. We consider a periodic boundary condition in the $y$-direction and introduce the Fourier transformation (FT):

$$\hat{c}_{j,k_y} = \sum_{m=1}^{M} e^{-ik_y m}\hat{c}_{j,m}, \quad \hat{c}^\dagger_{j,k_y} = \sum_{m=1}^{M} e^{ik_y m}\hat{c}^\dagger_{j,m}. \qquad (9)$$

Then, the Hamiltonian transforms to $H_{\mathrm{IQH}} = \sum_{k_y} H_{k_y}$, with $k_y$ being the quasi-momentum in the $y$-direction, where

$$H_{k_y} = t_x \sum_j^{N-1} (\hat{c}^\dagger_{j,k_y}\hat{c}_{j+1,k_y} + \mathrm{H.c.})$$
$$+ 2t_y \sum_n^{N} \cos(2\pi bj + k_y)\hat{c}^\dagger_{j,k_y}\hat{c}_{j,k_y}. \qquad (10)$$

By replacing $t_x$, $2t_y$, and $q$ with $J$, $\Delta$, and $\phi$, we obtain the 1D Aubry-André-Harper (AAH) model

$$H_{\mathrm{AAH}} = J \sum_j^{N-1} (\hat{c}^\dagger_j\hat{c}_{j+1} + \mathrm{H.c.}) + \Delta \sum_j^{N} \cos(2\pi bj + \phi)\hat{c}^\dagger_j\hat{c}_j, \qquad (11)$$

where the second index has been omitted in the 15-qubit experiment.

Since we simulated a 1D tight-binding fermionic Hamiltonian (11) in our experiments, to investigate the behaviour of one excitation (a hard-core boson) effectively captures the topological property of the system. Furthermore, in the case when we only excite one qubit, the interaction term in equation (4) is zero, and the behaviour of the quantum system we simulated is not affected by the statistics of the particle (fermionic or bosonic). Since our device is designed to fulfil the hard-core limit $|\eta/J| \simeq 25 \gg 1$, the highly occupied states of

transmon qubits are blockaded, which represents fermionisation of strongly interacting bosons[30]. Thus, the dynamical behaviour of a multiple-excitation system in the 15-qubit-chain experiment is similar as a 1D fermionic system, as demonstrated in ref. 30.

## Band structure spectroscopy and effect of decoherence

After initialising the selected qubits at their idle points, we prepare one target qubit $Q_{j,s}$ in the superposed state $|+_{j,s}\rangle = (|0_{j,s}\rangle + |1_{j,s}\rangle)/\sqrt{2}$, using a $Y_{\frac{\pi}{2}}$ pulse. Then, all qubits are tuned to their corresponding frequencies for the quench dynamics, and at a time $t$, we measure the $Q_{j,s}$ at its idle point in the $\hat{\sigma}^x$ and $\hat{\sigma}^y$ bases. For each $\phi$, time evolutions of $\langle \hat{\sigma}^x_{j,s}(t)\rangle$ and $\langle \hat{\sigma}^y_{j,s}(t)\rangle$ are monitored. Then, we calculate the squared Fourier transformation (FT) magnitude $|A_{j,s}|^2$ of the response function[28]

$$\chi_{j,s}(t) \equiv \langle \hat{\sigma}^x_{j,s}(t)\rangle + i\langle \hat{\sigma}^y_{j,s}(t)\rangle. \tag{12}$$

With the summation of the squared FT magnitudes of all selected qubits $I_\phi \equiv \sum_{j,s}|A_{j,s}|^2$, the positions of its peaks clearly indicate the eigenenergies $\{E_n\}$ of the system for each $\phi$.

Given $|\Psi_{j,s}(0)\rangle = \sum_n c_n^{(j,s)}|\psi_n\rangle$, with $H|\psi_n\rangle = E_n|\psi_n\rangle$ and $c_n^{(j,s)} \equiv \langle \psi_n|\Psi_{j,s}(0)\rangle$, and the evolution of the state can be obtained by solving the Schrödinger equation as

$$|\Psi_{j,s}(t)\rangle = e^{-iHt}|\Psi_{j,s}(0)\rangle = \sum_n c_n^{(j,s)} e^{-itE_n}|\psi_n\rangle. \tag{13}$$

We obtain the response function of the system after the initial perturbation as

$$\chi_{j,s}(t) \sim 2\langle \Psi_{j,s}(0)|\Psi_{j,s}(t)\rangle - 1 = 2\sum_n |c_n^{(j,s)}|^2 e^{-itE_n} - 1, \tag{14}$$

and the FT of the response function is calculated as

$$\begin{aligned}\tilde{\chi}_{j,s}(\omega) &\sim \frac{1}{2\pi}\int d\omega \left[ 2\sum_n |c_n^{(j,s)}|^2 e^{-it(E_n - \omega)} - e^{i\omega t}\right] \\ &= 2\sum_n |c_n^{(j,s)}|^2 \delta(\omega - E_n) - \delta(\omega),\end{aligned} \tag{15}$$

where the eigenenergies $\{E_n\}$ are indicated by the peaks of the FT signals.

Furthermore, when considering decoherence with a decaying rate $\gamma$ in a form

$$\begin{aligned}\chi_{j,s}(t) &\sim 2\langle \Psi_{j,s}(0)|\Psi_{j,s}(t)\rangle - 1 \\ &= \left[ 2\sum_n |c_n^{(j,s)}|^2 \delta(\omega - E_n) - \delta(\omega)\right] e^{-\gamma t},\end{aligned} \tag{16}$$

the FT of the response function is calculated as

$$\begin{aligned}\tilde{\chi}_{j,s}(\omega) &\sim \frac{1}{2\pi}\int d\omega \sum_n |c_n^{(j,s)}|^2 e^{-it(E_n - \omega)} \\ &= \sum_n |c_n^{(j,s)}|^2 \frac{4\gamma}{\gamma^2 + (\omega - E_n)^2} - \frac{2\gamma}{\gamma^2 + \omega^2},\end{aligned} \tag{17}$$

which indicates that the presence of decoherence increases the width of the peaks of the FT signals, and the locations of the peaks can still indicate the values of the eigenenergies.

There also exists the unwanted zero frequency signal of the FT. In our experiments, to eliminate the unwanted zero frequency signal of the FT we calculate the FT of oscillations of the response function $\delta\chi(t) \equiv \chi(t) - \overline{\chi(t)}$, i.e., the response function minus the its average over the time interval.

## Avoiding rung-pair excitations in strongly interacting Bose-Hubbard ladders

In the 30-qubit-ladder experiment, we excited either one corner qubit or a bulk qubit and monitored the quantum walks of the excitation, to study the topologically protected edge states of the bilayer quantum Hall systems. Localisation of the excitation initialised at a corner qubit and the propagation of the excitation initialised at the bulk qubit indicate the existence of a topological edge state protected by the topology of the bulk structure.

However, we do not consider to excite two corner qubits at the same edge simultaneously, because it has been experimentally and theoretically shown in refs. 29,50 that the dynamics of single- and double-excitation states have very distinct behaviours. Specifically, in the hard-core limit, there exists rung-pair localisation at the edges even for the topologically trivial case without the modulation of the on-site potentials, $\Delta_\uparrow/2\pi = \Delta_\downarrow/2\pi = 12$ MHz. In the centre-of-mass frame, the two-particle system can be mapped into an effective single-particle Hamiltonian, and there exists a zero-energy flat band in the hard-core limit, which is the origin of the localisation[50]. Therefore, in the 30-qubit-ladder experiment, we avoid exciting two qubits on the same rung simultaneously, when investigating the topologically protected edge states.

## Quantum charge pumping

In addition to the study of 2D topological systems, topological charge pumping provides an alternative way to explore the quantised transport with topological protection in a dynamical 1D system. The concept of topological charge pumping was first proposed by Thouless[10], and recognised a topological quantisation of charge transport in a 1D time-varying potential driven in adiabatic cycles. The charge transported in a pumping cycle is determined by the Chern number[4], which is defined over a 2D Brillouin zone with one spatial dimension and one temporal dimension.

We experimentally simulate the quantum charge pump by adiabatically varying $\phi$ in a $2\pi$ period starting from $\phi_0 = \frac{5\pi}{3}$ with $\Delta/2\pi = 36$ MHz. After initialising 15 qubits in the state $|0\rangle^{\otimes 15}$ at their idle frequencies, we prepared the central qubit, $Q_{8,\uparrow}$, in the $|1\rangle$ state. Then, we set the frequencies of qubits as $\omega_j(\phi) = \omega_0 + \Delta \sum_{j=1}^{15} \cos(2\pi bj + \phi)$, with $\Delta/2\pi = 36$ MHz and an initial phase $\phi_0 = \frac{5\pi}{3}$. In this case, the single-excitation initial state has the minimum energy, and with a high value of $\Delta$, the initial excitation stabilises at $Q_{8,\uparrow}$.

The frequencies for all qubits are calibrated using the frequency calibration procedure as shown in the Supplementary Information. Then, by modulating the frequencies of all 15 qubits simultaneously, we slowly vary $\phi$ from $\phi_0$ to $\phi_0 \pm 2\pi$ in 1100 ns with a relatively slow speed -1.8$\pi$/μs for the backward and forward pumping schemes, respectively. For the case of no pump, $\phi$ is fixed at $\phi_0$ during the time evolution.

In our experiments, we measured the $|1\rangle$-state occupation probability of each qubit at its idle point in the $\hat{\sigma}^z$ basis for each evolution time $t$. To reduce the effect of the stochastic fluctuations, we maintained a fixed sample of 4000 single-shot readouts and repeated the measurement procedure 10 times for estimating the mean values and standard deviations at each evolution time $t$. We obtained the average displacements of the centre of mass (CoM) as

$$\delta x(t) = \sum_{j=1}^{15} P_{j,\uparrow}(t)(j - 8)/3, \tag{18}$$

where $P_j(t)$ is the $|1\rangle$-state occupation probability of the $Q_{j,\uparrow}$. Note that the CoM is divided by 3, because we set $b = \frac{1}{3}$ and there are 3 qubits in one unit cell. The experimental results can be found in Fig. 3 in the main text.

## Characterisation of bilayer Chern insulators

The Hamiltonian of the 30-qubit ladder reads

$$
\begin{aligned}
H = \sum_{j,s} & (J_\parallel \hat{c}^\dagger_{j,s}\hat{c}_{j+1,s} + J'_\parallel \hat{c}^\dagger_{j,s}\hat{c}_{j+2,s} + \text{H.c.}) \\
& + J_\perp \sum_j (\hat{c}^\dagger_{j,\uparrow}\hat{c}_{j,\downarrow} + \text{H.c.}) \\
& + J_\times \sum_j (\hat{c}^\dagger_{j,\uparrow}\hat{c}_{j+1,\downarrow} + \hat{c}^\dagger_{j,\downarrow}\hat{c}_{j+1,\uparrow} + \text{H.c.}) \\
& + \sum_{j,s} \Delta_s \cos(2\pi b_s j + \phi)\hat{c}^\dagger_{j,s}\hat{c}_{j,s},
\end{aligned}
\tag{19}
$$

where $s \in \{\uparrow, \downarrow\}$, and $b = \frac{1}{3}$ determines the modulation periodicity. The typical hopping strengths for our sample are $J_\parallel/2\pi = 8$ MHz, $J_\perp/2\pi = 7$ MHz, $J'_\parallel \simeq 0.1 J_\parallel$, and $J_\times \simeq 0.2 J_\parallel$. The AAH ladder can be mapped into two coupled Hofstadter lattices, subjected to the same effective magnetic fields for each layer.

We now proceed to discuss the topological properties in two cases, i.e.,

$$
\Delta_\uparrow = \Delta_\downarrow = \Delta, \text{ and } \Delta_\uparrow = -\Delta_\downarrow = \Delta.
$$

They correspond to the study of two types of bilayer quantum systems with two layers having the same magnetic flux and a $\pi$-flux difference, respectively. In our experiments, we set $\Delta/2\pi = 12$ MHz.

### Two qubit chains with the same periodically modulated on-site potentials

For $\Delta_\uparrow = \Delta_\downarrow = \Delta$, two identical AAH chains are coupled to form an AAH ladder. In Fig. 5, we plot the band structures for different inter-chain hopping strengths $J_\perp$. Figure 5 shows the energy spectra of different topological phase regimes, showing that topological phase transitions occur as $J_\perp$ varies.

We can identify the topologically nontrivial and trivial bands by using the Chern number of each band[4], which is defined as

$$
C_n = \frac{1}{2\pi}\int_0^{2\pi} dk \int_0^{2\pi} d\phi \left[\partial_k \mathcal{A}^n_\phi - \partial_\phi \mathcal{A}^n_k\right].
\tag{20}
$$

The $n$th band Berry connection $\mathcal{A}^n$ is written as

$$
\mathcal{A}^n_\gamma = i\langle\varphi_n(k,\phi)|\partial_\gamma|\varphi_n(k,\phi)\rangle,
\tag{21}
$$

where $|\varphi_n(k,\phi)\rangle$ is the $n$th band's wavefunction, and $\gamma = k, \phi$. Then, the Hall conductivity reads $\sigma = \sum_n C_n$, with $n$ being summed over the occupied bands.

When the inter-chain hopping strength $J_\perp$ is much smaller (Fig. 5a) or much larger (Fig. 5c) than the intra-chain hoping strength $J_\parallel$, the topological boundary states are well characterised by the Chern number. However, when $J_\perp$ is comparable to $J_\parallel$, topological edge states with zero Hall conductivity appear (Fig. 5b) for the half filling.

The zero Hall conductivity results from the contribution of a pair of counter-propagating chiral edge states. This novel topological phase was also studied in a dimerised Hofstadter model[44] and has never been experimentally observed before. Figure 6 shows the

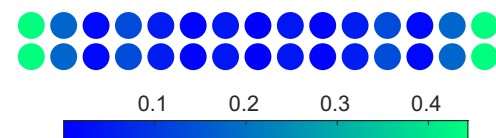

**Fig. 6 | Mid-gap state for the bilayer topological system with the same periodically modulated on-site potentials.** Density distribution of a mid-gap state for $\Delta_\uparrow/2\pi = \Delta_\downarrow/2\pi = 12$ MHz and $\phi = \frac{2\pi}{3}$ with $J_\perp/2\pi = 7$MHz, $J_\parallel/2\pi = 8$ MHz, $J'_\parallel = 0.1 J_\parallel$, and $J_\times = 0.2 J_\parallel$.

density distribution of a mid-gap state for the half filling, where the in-gap state occupies the end sites of both chains.

The topological edges with zero Chern number (i.e., zero Hall conductivity) is protected by the inversion symmetry. After Fourier transforming Eq. (19), we have the following momentum-space Hamiltonian

$$
H(k) = \sum_{\mathbf{k}} \Phi^\dagger_{\mathbf{k}}
\begin{pmatrix}
\Delta\cos(\frac{2\pi}{3}+\phi) & J_\parallel + J'_\parallel e^{-ik} & J_\parallel e^{-ik}+J'_\parallel & J_\perp & J_\times & J_\times e^{-ik} \\
J_\parallel + J'_\parallel e^{ik} & \Delta\cos(\frac{4\pi}{3}+\phi) & J_\parallel + J'_\parallel e^{-ik} & J_\times & J_\perp & J_\times \\
J_\parallel e^{ik}+J'_\parallel & J_\parallel + J'_\parallel e^{ik} & \Delta\cos(\phi) & J_\times e^{ik} & J_\times & J_\perp \\
J_\perp & J_\times & J_\times e^{-ik} & \Delta\cos(\frac{2\pi}{3}+\phi) & J_\parallel + J'_\parallel e^{-ik} & J_\parallel e^{-ik}+J'_\parallel \\
J_\times & J_\perp & J_\times & J_\parallel + J'_\parallel e^{ik} & \Delta\cos(\frac{4\pi}{3}+\phi) & J_\parallel + J'_\parallel e^{-ik} \\
J_\times e^{ik} & J_\times & J_\perp & J_\parallel e^{ik}+J'_\parallel & J_\parallel + J'_\parallel e^{ik} & \Delta\cos(\phi)
\end{pmatrix}
\Phi_{\mathbf{k}},
\tag{22}
$$

where $\Phi_{\mathbf{k}} = (\hat{c}_{k,1,\uparrow}, \hat{c}_{k,2,\uparrow}, \hat{c}_{k,3,\uparrow}, \hat{c}_{k,1,\downarrow}, \hat{c}_{k,2,\downarrow}, \hat{c}_{k,3,\downarrow})^T$, with $\hat{c}_{k,l,s}$ ($l = 1, 2, 3$ and $s \in \{\uparrow, \downarrow\}$) being the annihilation operator of a hardcore boson at momentum $k$, sublattice $l$, and pesudospin $s$.

The Hamiltonian $H(k)$ in Eq. (22) has the inversion symmetry

$$
\hat{\mathcal{P}}H(k)\hat{\mathcal{P}}^{-1} = H(-k),
\tag{23}
$$

for $\phi = \frac{2\pi}{3}$ and $\phi = \frac{5\pi}{3}$, where the inversion symmetry operator $\hat{\mathcal{P}}$ is

$$
\hat{\mathcal{P}} =
\begin{pmatrix}
0 & 0 & 0 & 0 & 0 & 1 \\
0 & 0 & 0 & 0 & 1 & 0 \\
0 & 0 & 0 & 1 & 0 & 0 \\
0 & 0 & 1 & 0 & 0 & 0 \\
0 & 1 & 0 & 0 & 0 & 0 \\
1 & 0 & 0 & 0 & 0 & 0
\end{pmatrix}.
\tag{24}
$$

Then, we define an integer invariant $\mathcal{N}$ to characterize this novel topological phase, which is expressed as[44]

$$
\mathcal{N} \equiv |N_1 - N_2|,
\tag{25}
$$

where $N_1$ and $N_2$ are the number of negative parities (by applying the inversion symmetry operator to eigenstates) at the high symmetry points $k = 0$ and $k = \pi$, respectively.

Figure 7 shows the Bloch band structures for $\phi = \frac{2\pi}{3}$ and $\phi = \frac{5\pi}{3}$, respectively, and the topological invariant $\mathcal{N}$ at each band gap is indicated. For $\phi = \frac{2\pi}{3}$, we find $\mathcal{N} = 1$ for $\frac{1}{4}$-filling and half-filling, indicating a pair of edge states pinned at $\phi = \frac{2\pi}{3}$. For $\phi = \frac{5\pi}{3}$, we have $\mathcal{N} = 1$ for $\frac{3}{4}$-filling and half-filling, indicating a pair of edge states pined at $\phi = \frac{5\pi}{3}$. Therefore, two pairs of edge states appear at half-filling, which propagate in opposite directions at each edge.

### Two qubit chains with opposite periodically modulated on-site potentials

For $\Delta_\uparrow = -\Delta_\downarrow = \Delta$, the AAH ladder shows two typical topological band structures, as shown in Fig. 8. This bilayer structure, with two layers

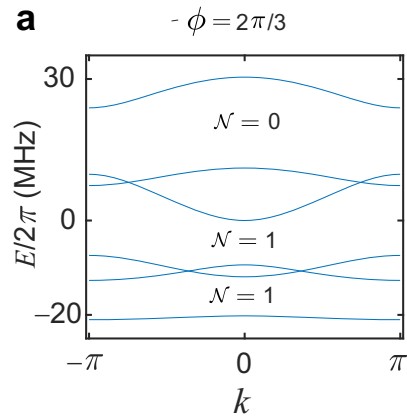
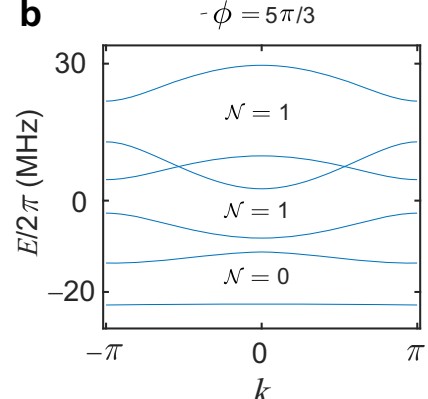

**Fig. 7 | Bloch energy bands for the bilayer topological system with the same periodically modulated on-site potentials. a**, **b** Bloch energy bands versus $k$ for identical chains in a qubit ladder, with modulated amplitudes $\Delta_\uparrow/2\pi = \Delta_\downarrow/2\pi = 12$ MHz with $\phi = \frac{2\pi}{3}$ (**a**), and $\phi = \frac{5\pi}{3}$ (**b**). The topological invariant $\mathcal{N}$ at each band gap is indicated, and $\mathcal{N} = 1$ indicates a pair of edge states. Other parameters used are $J_\perp/2\pi = 7$ MHz, $J_\parallel/2\pi = 8$ MHz, $J'_\parallel = 0.1J_\parallel$, and $J_\times = 0.2J_\parallel$.

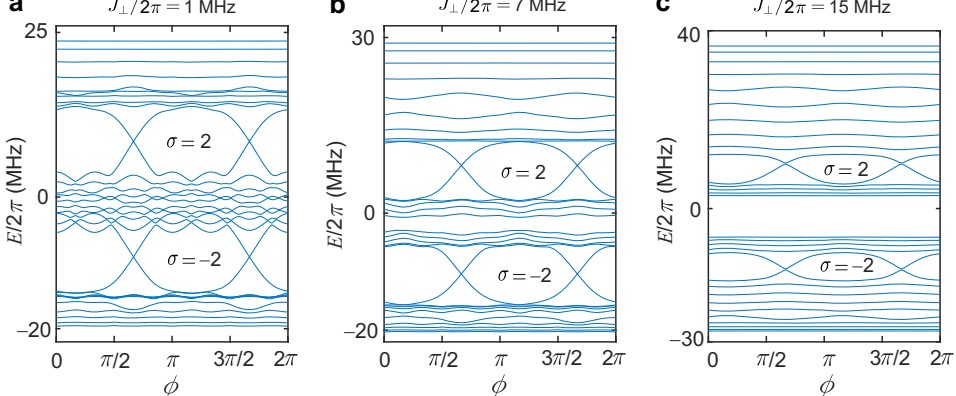

**Fig. 8 | Energy spectra for the bilayer topological system with opposite periodically modulated on-site potentials. a**–**c** Energy spectra versus $\phi$ of a finite ladder with 30 sites for opposite modulated amplitudes $\Delta_\uparrow/2\pi = -\Delta_\downarrow/2\pi = 12$ MHz with $J_\perp/2\pi = 1$ MHz (**a**), $J_\perp/2\pi = 7$ MHz (**b**), and $J_\perp/2\pi = 15$ MHz (**c**). Other parameters used are $J_\parallel/2\pi = 8$ MHz, $J'_\parallel = 0.1J_\parallel$, and $J_\times = 0.2J_\parallel$.

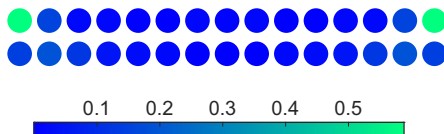
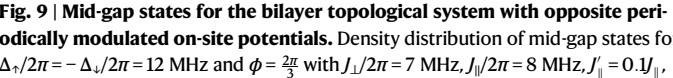
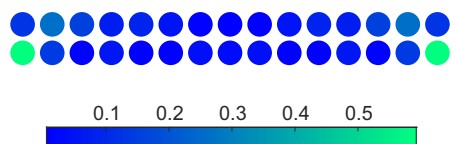

**Fig. 9 | Mid-gap states for the bilayer topological system with opposite periodically modulated on-site potentials.** Density distribution of mid-gap states for $\Delta_\uparrow/2\pi = -\Delta_\downarrow/2\pi = 12$ MHz and $\phi = \frac{2\pi}{3}$ with $J_\perp/2\pi = 7$ MHz, $J_\parallel/2\pi = 8$ MHz, $J'_\parallel = 0.1J_\parallel$, and $J_\times = 0.2J_\parallel$. The left panel is for a mid-gap state in the second gap; and the right panel is for a mid-gap state in the fourth gap. Note that the first and fifth gaps in Fig. 8b are not clearly shown due to finite-size effects.

having a $\pi$-flux difference, shows different topological features from the bilayer structure with two layers having the same magnetic flux. Furthermore, the in-gap states mainly occupy the end site of one of the chains, as shown in Fig. 9. Thus, this synthetic bilayer quantum system, with two layers having a $\pi$-flux difference, presents an integer quantum Hall effect with chiral edge states, identified by higher Chern numbers.

## Data availability

The source data underlying all figures are available at https://doi.org/10.6084/m9.figshare.23925009. Other data are available from the corresponding author upon request.

## Code availability

The codes are available upon reasonable request from the corresponding author.

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

## Acknowledgements

We thank Hongming Weng for stimulating discussions. This research was supported by the NSFC (Grants No. T2121001, No. 11934018, No. 11904393, No. 92065114, No. 12174207, No. 12274142), the CAS Strategic Priority Research Program (Grant No. XDB28000000, No. YJKYYQ20200041), the Beijing Natural Science Foundation (Grant No. Z200009), the State Key Development Program for Basic Research of China (Grant No. 2017YFA0304300), the Key-Area Research and Development Program of Guangdong Province China (Grant No. 2020B0303030001). F.N. is supported in part by: Nippon Telegraph and Telephone Corporation (NTT) Research, the Japan Science and Technology Agency (JST) [via the Quantum Leap Flagship Program (Q-LEAP), and the Moonshot R&D Grant Number JPMJMS2061], the Asian Office of Aerospace Research and Development (AOARD) (via Grant No. FA2386-20-1-4069), and the Foundational Questions Institute Fund (FQXi) via Grant No. FQXi-IAF19-06.

## Author contributions

Y.-R.Z., Tao Liu, K.X., F.N., and H.F. conceived the research; Y.-R.Z., Tao Liu, Z.-C.X., K.H., K.X., F.N., and H.F. designed the experiment; Z.-C.X. designed and fabricated the device with the help of G.-H.L., Z.-Y.M., X.S., and D.Z.; K.H. and C.-L.D performed the experiment supervised by K.X.,

Z.-B.L., and H.F.; Tong Liu, H.L and Y.T. helped to build up the experimental setup supervised by K.X.; K.H., Y.-R.Z., and Y.-H.S. performed numerical simulations; K.H., Y.-R.Z., and Z.-C.X. analysed experimental results. Tao Liu; Y.-R.Z., F.N., and H.F. performed theoretical explanations; G.X. and H.Y. provide the Josephson Parametric Amplifiers; All authors contributed to the discussions of the results and the development of the manuscript; F.N. and H.F. supervised the whole project.

## Competing interests

The authors declare no competing interests.
