## [Peer Review File · Nature Communications]

REVIEWER COMMENTS

Reviewer #1 (Remarks to the Author):

This work demonstrates the simulation of various topological phases based on a collection of transmon qubits.

The quality of the data is very impressive, it proves that superconducting processors are extremely versatile tools to explore quantum systems as coined by Feynmann 40 years ago. The extra dimension synthesis is quite neat. The level of experimental control is astonishing.

The method section details the underlying theory details, for instance, the hard-core boson limit and its relation with transmon physics.

The supplementary material is extremely instructive about the calibration of each qubit, the coupling maps, and many other experimental details that makes this experiment reproducible by highly skilled experimentalists.

The main question I have is that it seems that only single-particle dynamics are explored in this work. A section in the method mentions rung-pair excitations that should be avoided. Are there any non-trivial many-body dynamics in this experiment that could be explored? I'm not asking for further experimental data but for a qualitative understanding of many-body effects.

Anyway, this work is one of the best quantum simulation with circuits I have seen, I recommend this work for publication.

Congratulations to the authors for this beautiful piece of physics.

Reviewer #2 (Remarks to the Author):

In this manuscript [NCOMMS-22-37190-T], Xiang et al. design, fabricate, probe, and measure a superconducting circuit device to access 2D and bilayer integer quantum Hall systems. They simulate a 2D square lattice with periodic boundary conditions in one direction and length 15 sites along the other, threaded by $1/3$ flux quantum, via its mapping to the Aubry-André-Harper (AAH) model. For this system, they map out the band structure and demonstrate dynamical localization of edge states and topological charge pumping. They conclude with analogous measurements for a pair of coupled 1D AAH chains, simulating a bilayer Chern insulator.

To me, this work seems like an extension of ref. 34 (Roushan et al., *Science* 358 1175 (2017)). To begin, the 1D AAH chain here is 15 sites whereas in ref. 34 it was 9 sites. In this work, the authors vary the modulation phase in the AAH model to access different quasimomenta for the 2D Chern insulator band spectrum, rather than vary the modulation periodicity as was done in ref. 34. However, the time-domain spectroscopy technique to extract eigenenergies seems the same.

This work additionally examines the dynamics of edge states and topological pumping, plus extends their measurements to the pair of coupled AAH chains, which differ more substantially from the results of ref. 34. At the same time, it is not immediately clear to me how these results demonstrate additional capabilities for “studying different intriguing topological phases of quantum matter” as in claimed at the end of the abstract and at the end of the introduction. In particular, these simulations are fundamentally limited to single-particle physics and to models that can be reduced to an effective 1D model. The paper could be strengthened significantly if the authors more explicitly suggested areas of quantum simulation which are now more immediately accessible as a result of their work.

Having said that, it seems that the work does otherwise support the authors’ conclusions and claims. The methodology and data quality largely appear sound, as well as the data analysis and interpretations. I do find it peculiar that the authors found such large deviations from the Chern numbers ± 1 in Figure 3d, and was wondering if the authors could perform a more quantitative and systematic analysis around the proposed origins of the deviations (boundary reflection, fast pumps, and higher-band population). There indeed seems to be enough detail provided in the methods to reproduce this experiment.

Reply to Report of Reviewer #1 – NCOMMS-22-37190-T

(1-0) REVIEWER 1 WROTE THAT:

This work demonstrates the simulation of various topological phases based on a collection of transmon qubits.

The quality of the data is very impressive, it proves that superconducting processors are extremely versatile tools to explore quantum systems as coined by Feynmann 40 years ago. The extra dimension synthesis is quite neat. The level of experimental control is astonishing.

The method section details the underlying theory details, for instance, the hard-core boson limit and its relation with transmon physics.

The supplementary material is extremely instructive about the calibration of each qubit, the coupling maps, and many other experimental details that makes this experiment reproducible by highly skilled experimentalists.

OUR REPLY:

We thank Reviewer 1 for the recognition of the correctness, the quality, and the impact of our manuscript.

(1-1) REVIEWER 1 WROTE THAT:

The main question I have is that it seems that only single-particle dynamics are explored in this work. A section in the method mentions rung-pair excitations that should be avoided. Are there any non-trivial many-body dynamics in this experiment that could be explored? I'm not asking for further experimental data but for a qualitative understanding of many-body effects.

OUR REPLY:

We thank Reviewer 1 for pointing out this issue. In this work, we simulate various types of Chern insulators, which can be completely investigated by observing single excitations of a qubit. Since our system is described by the Bose-Hubbard model in the hard-core limit, $U/J \gg 1$, the cases for the qubit chain and qubit ladder become very different.

For the qubit chain, the hard-core boson behaves like fermions, as we have shown in [Yan2019]. Therefore, single-particle dynamics is sufficient to analyse Chern insulators, and

multi-particle dynamics does not provide further insights into this problem. To further investigate the many-body effects, we need to design new superconducting samples, whose hopping strength J and on-site nonlinear interaction strength U were tuneable. Then, we could simulate some non-trivial many-body dynamics for investigating the many-body physics of topological insulators with interactions, which extends the topics of this manuscript. For example, we could investigate the many-body physics of the Bose-Mott insulators using the superconducting qubit chain with periodic superlattice potentials as proposed in Ref. [Zhu2013]. For this many-body physics, this many-body system shows topological phases at $1/3$ filling factor and $2/3$ filling factor, of which the topological edge states can be simulated and observed by edge excitation after $1/3$ or $2/3$ filling.

For the qubit ladder, as discussed in Methods, we did not consider to excite two corner qubits at the same edge simultaneously, because it has been experimentally and theoretically shown in Refs. [Ye2019, Li2020] that the dynamics of single- and double-excitation states behave very differently. In the hard-core limit, $U/J \gg 1$, there exists rung-pair localisation at the edges even for the topologically trivial case without the modulation of the on-site potentials. Thus, to excite multiple qubits would be not suitable for simulating bilayer Chern insulators. Nevertheless, we agree that nontrivial many-body physics could be investigated in superconducting qubit ladders with more controllability in addition to localisation of rung pairing excitations [Ye2019, Li2020], e.g., the hole pairing in ladder systems [Hirthe2023]. All of these projects will be the subject of future research work.

In the Section Conclusion and Discussion, we have added several sentences discussing the possible demonstration of quantum simulating different intriguing topological physics with updated superconducting processor:

“For instance, by upgrading the qubit-ladder superconducting processor with tuneable interactions between two qubit chains, we could observe the quantum phase transitions between topological phases with zero Chern number (Extended Data Fig. 1b) and with higher Chern numbers (Extended Data Fig. 1c). In addition, when tuning weak couplings between two chains and modulating b either positive or negative for two chains, respectively, the quantum spin Hall effect could be demonstrated by simulating bilayer quantum Hall systems with opposite Chern numbers [Kane2005]. In addition, quantum many-body physics could be experimentally studied on the qubit-ladder superconduct-

ing processor with more controllability, e.g., Bose-Mott insulators [Zhu2013] and hole pairing in ladder systems [Hirthe2023].”

- [Yan2019] Z. Yan, *et al.* “Strongly correlated quantum walks with a 12-qubit superconducting processor”, *Science* **364**, 753 (2019).
- [Zhu2013] S. L. Zhu, *et al.* “Topological Bose-Mott insulators in a one-dimensional optical superlattice”, *Phys. Rev. Lett.* **110**, 075303 (2013).
- [Ye2019] Y. Ye, *et al.* “Propagation and localization of collective excitations on a 24-qubit superconducting processor”, *Phys. Rev. Lett.* **123**, 050502 (2019).
- [Li2020] S.-S. Li, Z.-Y. Ge, and H. Fan, “Localization of rung pairs in a hard-core Bose-Hubbard ladder”, *Phys. Rev. A* **102**, 062409 (2020).
- [Hirthe2023] S. Hirthe, *et al.* “Magnetically mediated hole pairing in fermionic ladders of ultracold atoms”, *Nature* **613**, 463 (2023).

(1-2) REVIEWER 1 WROTE THAT:

Anyway, this work is one of the best quantum simulation with circuits I have seen, I recommend this work for publication.

Congratulations to the authors for this beautiful piece of physics.

OUR REPLY:

We thank Reviewer 1 for writing “*I recommend this work for publication*”. We hope that with these revisions and explanations, our manuscript is now suitable for publication in *Nature Communications*.

Reply to Report of Reviewer #2 – NCOMMS-22-37190-T

(2-0) REVIEWER 2 WROTE THAT:

In this manuscript [NCOMMS-22-37190-T], Xiang *et al.* design, fabricate, probe, and measure a superconducting circuit device to access 2D and bilayer integer quantum Hall systems. They simulate a 2D square lattice with periodic boundary conditions in one direction and length 15 sites along the other, threaded by $1/3$ flux quantum, via its mapping to the Aubry-André-Harper (AAH) model. For this system, they map out the band structure and demonstrate dynamical localization of edge states and topological charge pumping. They conclude with analogous measurements for a pair of coupled 1D AAH chains, simulating a bilayer Chern insulator.

OUR REPLY:

We thank Reviewer 2 for careful reading our manuscript, the recognition of the correctness of our results, and the comments that help a lot to improve the impact of our manuscript. We are enclosing the new version of our paper revised according to all comments, where the changes are marked in red in the additional manuscript.

(2-1) REVIEWER 2 WROTE THAT:

To me, this work seems like an extension of ref. 34 (Roushan *et al.*, Science 358 1175 (2017)). To begin, the 1D AAH chain here is 15 sites whereas in ref. 34 it was 9 sites. In this work, the authors vary the modulation phase in the AAH model to access different quasimomenta for the 2D Chern insulator band spectrum, rather than vary the modulation periodicity as was done in ref. 34. However, the time-domain spectroscopy technique to extract eigenenergies seems the same. This work additionally examines the dynamics of edge states and topological pumping, plus extends their measurements to the pair of coupled AAH chains, which differ more substantially from the results of ref. 34.

OUR REPLY:

We agree with Reviewer 2 that in this work we have used similar techniques as in Refs. [34] to extract the energy band structure of the simulated topological systems. Moreover, together with the measured band structure and the dynamics of edge excitation, our work

experimentally demonstrates the bulk-edge correspondence in several types of Chern insulators, which is an important result. In addition, topological charge pumping was demonstrated to investigate the Chern insulators.

(2-2) REVIEWER 2 WROTE THAT:

At the same time, it is not immediately clear to me how these results demonstrate additional capabilities for “studying different intriguing topological phases of quantum matter” as in claimed at the end of the abstract and at the end of the introduction. In particular, these simulations are fundamentally limited to single-particle physics and to models that can be reduced to an effective 1D model. The paper could be strengthened significantly if the authors more explicitly suggested areas of quantum simulation which are now more immediately accessible as a result of their work.

OUR REPLY:

We thank Reviewer 2 for the comments.

We first discuss the potential capabilities of the ladder structure for investigating different intriguing topological phases of quantum matter. First, if the hopping strengths between two qubit chain were tuneable, we could observe the quantum phase transitions between the topological phase with zero Chern number (investigated in our manuscript), and another type of Chern insulators with higher Chern numbers, as shown in Extended Data Fig. 1(b) and 1(c) for the band structures with different hopping strengths. This transition could be observed by increasing the hopping strengths between two qubit chains. Since high-Chern-number insulators without the formation of Landau levels have attracted increasing attention [Ge2020,Chen2020,Zhao2020], our work provides a new perspective for exploring these emergent topological phases.

Second, we could also simulate the quantum spin Hall effect on the qubit-ladder sample, when tuning the couplings between two chains as very weak and tuning b either positive or negative (simulating magnetic fluxes with different directions) for two chains, respectively. This scheme is very similar to the early proposal by Kane and Mele [Kane2005] for realising quantum spin Hall effect with two weakly coupled Haldane models with opposite Chern numbers.

In addition, we could also consider to investigate the many-body physics on the ladder

sample when the model is not in the hard-core limit. This could be realised when the hopping between qubits is tuneable by adding couplers. For example, we could investigate the many-body physics of Bose-Mott insulators using the superconducting qubit chain with periodic superlattice potentials as proposed in Ref. [Zhu2013]. For this many-body physics, this many-body system shows topological phases at $1/3$ filling factor and $2/3$ filling factor, of which the topological edge states can be simulated and observed by edge excitation after $1/3$ or $2/3$ filling. Also, besides localisation of rung pairing excitations [Ye2019, Li2020] in Methods, we could also investigate nontrivial many-body physics e.g., the hole pairing in ladder systems [Hirthe2023] with more controllability of the superconducting circuits. These extensions correspond to future work.

In the Section Conclusion and Discussion, we have added several sentences discussing the possible future study of quantum simulating different intriguing topological physics with upgraded ladder-type superconducting processor:

“For instance, by upgrading the qubit-ladder superconducting processor with tuneable interactions between two qubit chains, we could observe the quantum phase transitions between topological phases with zero Chern number (Extended Data Fig. 1b) and with higher Chern numbers (Extended Data Fig. 1c). In addition, when tuning weak couplings between two chains and modulating b either positive or negative for two chains, respectively, the quantum spin Hall effect could be demonstrated by simulating bilayer quantum Hall systems with opposite Chern numbers [Kane2005]. In addition, quantum many-body physics could be experimentally studied on the qubit-ladder superconducting processor with more controllability, e.g., Bose-Mott insulators [Zhu2013] and hole pairing in ladder systems [Hirthe2023].”

[Kane2005] C. L. Kane and E. J. Mele, “Quantum spin Hall effect in graphene”, Phys. Rev. Lett. **95**, 226801 (2005).

[Ge2020] J. Ge, *et al.* “High-Chern-number and high-temperature quantum Hall effect without Landau levels”, Natl. Sci. Rev. **7**, 1280 (2020).

[Chen2020] G. Chen, *et al.* “Tunable correlated Chern insulator and ferromagnetism in a moiré superlattice”, Nature **579**, 56 (2020).

- [Zhao2020] Y.-F. Zhao, *et al.* “Tuning the Chern number in quantum anomalous Hall insulators”, *Nature* **588**, 419 (2020).
- [Zhu2013] S. L. Zhu, *et al.* “Topological Bose-Mott insulators in a one-dimensional optical superlattice”, *Phys. Rev. Lett.* **110**, 075303 (2013).
- [Ye2019] Y. Ye, *et al.* “Propagation and localization of collective excitations on a 24-qubit superconducting processor”, *Phys. Rev. Lett.* **123**, 050502 (2019).
- [Li2020] S.-S. Li, Z.-Y. Ge, and H. Fan, “Localization of rung pairs in a hard-core Bose-Hubbard ladder”, *Phys. Rev. A* **102**, 062409 (2020).
- [Hirthe2023] S. Hirthe, *et al.* “Magnetically mediated hole pairing in fermionic ladders of ultracold atoms”, *Nature* **613**, 463 (2023).

(2-3) REVIEWER 2 WROTE THAT:

Having said that, it seems that the work does otherwise support the authors’ conclusions and claims. The methodology and data quality largely appear sound, as well as the data analysis and interpretations.

OUR REPLY:

We thank Reviewer 1 for the recognition of the correctness, the quality, and the correctness of our manuscript.

(2-4) REVIEWER 2 WROTE THAT:

I do find it peculiar that the authors found such large deviations from the Chern numbers ± 1 in Figure 3d, and was wondering if the authors could perform a more quantitative and systematic analysis around the proposed origins of the deviations (boundary reflection, fast pumps, and higher-band population). There indeed seems to be enough detail provided in the methods to reproduce this experiment.

OUR REPLY:

We thank Reviewer 2 for pointing out this issue, and we have also considered it to be an issue to re-examine. To completely address this, we have spent two months conducting the experiments with different parameters and verified that the deviation from Chern numbers mainly results from fast pumping.

FIG. R1: **a**, Experimental data (circles and squares for different periods $T = 410, 1,110$ ns, respectively) for the displacement of the centre of mass (CoM) δx versus time t within one pumping cycle, compared with the numerical simulations (solid curves) with different periods $T = 310, 410, 1,110$ ns. **b**, Numerical data for the displacement of the CoM δx in one pumping cycle versus different pumping periods T .

Additional experiments with a slower pumping rate and numerical simulations have been performed to investigate this problem, and the results are summarised in Fig. R1. In the new experiments, we have increased the period T from 410 ns to 1,110 ns, corresponding to a slower ramping speed of ϕ . The experimental results for the displacement of the centre of mass (CoM) δx versus the time t/T are compared in Fig. R1(a) with numerical simulations. It is clear in the revised Fig. 3d that the experimental data for $T = 1,110$ ns fit well with the Chern number ± 1 , and thus, we conclude that the deviations of our previous data from ± 1 mainly result from fast pumping. This can be also verified by the numerical results of the displacement of the CoM in one pumping cycle with different periods, as shown in Fig. R1(b). As the period is too short $T \lesssim 610$ ns, it presents fast pumping (in the fast pumping region); that is, large deviations from the Chern number $+1$, as the period becomes long enough, the data approximates the Chern number $+1$.

In the new version of our manuscript, we have replaced the data in Fig. 3(c1–c3,d) updated the experimental parameters in the main text with those with a period 1,110 ns, which is long enough for not being in the fast pumping region. In addition, we have also compared the old experimental data of fast pumping with the new data in the revised Supplementary

Information and give a brief discussion of the effect of fast pumping.

Summary of changes

All our changes are marked in red in the additional manuscript.

1. We have replaced the data in Fig. 3(c1–c3,d) updated the experimental parameters in the main text with those with new experimental parameters with a period 1,110 ns.
2. We have added a comparison between the old experimental data of fast pumping and the new data with a long period in the Supplementary Information and give a brief discussions about the effect of fast pumping.
3. In the conclusion part, we have added several sentences discussing the future potential of quantum simulating different intriguing topological phases with our superconducting quantum processor and its updated upgraded model. Three references [48,49,50] are added and cited here.
4. In Supplementary Information, have been added a subsection “Topological charge pumping and fast pumping”, discussing fast pumping according to the reply to the Comment (2-4). Two figures, Figs. S25 and S26 are added to show experimental and numerical data, comparing the topological charge pumping with fast pumping.

REVIEWERS' COMMENTS

Reviewer #2 (Remarks to the Author):

The authors have addressed my concerns in their revision. In particular, they have (1) understood and resolved the experimental discrepancy seen in Figure 3d, and (2) highlighted the potential significance of their work by connecting it more clearly to future studies.

However, the two studies mentioned in the last sentence of the conclusion require interaction strength tunability and fermion statistics, respectively. These are significantly more challenging to engineer than the qubit-qubit coupling tunability required for the other studies proposed. In light of this, I wonder if it may be worth explicitly mentioning that, or omitting the last sentence altogether, to not unintentionally mislead any readers.

Reply to REVIEWERS' COMMENTS

Reviewer #2 wrote that:

The authors have addressed my concerns in their revision. In particular, they have (1) understood and resolved the experimental discrepancy seen in Figure 3d, and (2) highlighted the potential significance of their work by connecting it more clearly to future studies.

However, the two studies mentioned in the last sentence of the conclusion require interaction strength tunability and fermion statistics, respectively. These are significantly more challenging to engineer than the qubit-qubit coupling tunability required for the other studies proposed. In light of this, I wonder if it may be worth explicitly mentioning that, or omitting the last sentence altogether, to not unintentionally mislead any readers.

Our reply:

We thank Reviewer #2 for this comment. To avoid any possible misunderstanding, we have omitted the last sentence, as suggested by Reviewer #2.